# Desktop Fabrication of Strong Poly (Lactic Acid) Parts: FFF Process Parameters Tuning

**DOI:** 10.3390/ma12132071

**Published:** 2019-06-27

**Authors:** Vladimir E. Kuznetsov, Azamat G. Tavitov, Oleg D. Urzhumtsev, Artem A. Korotkov, Sergey V. Solodov, Alexey N. Solonin

**Affiliations:** 1Department of Physical Metallurgy of Non-Ferrous Metals, National University of Science and Technology “MISIS”, Leninskiy Prospekt 4, NUST MISIS, 119049 Moscow, Russia; 2Departament of Automated Control Systems, National University of Science and Technology “MISIS”, Leninskiy Prospekt 4, NUST MISIS, 119049 Moscow, Russia

**Keywords:** fused filament fabrication, fused deposition modeling, interlayer bonding, technology optimization

## Abstract

The current study aims to evaluate the possibilities to increase part strength by optimizing the Fused Filament Fabrication (FFF) process parameters. Five different CAD models of parts with the same coupling dimensions but of different shape inherited from a recent study were converted into test samples with Ultimaker 2 3D printer. The main measure of success was the sample strength, defined as the load at which the first crack in the stressed area of the part appeared. Three different modifications to the FFF process with verified positive effect on interlayer bonding were applied. The first modification included raising the extrusion temperature and disabling printed part cooling. The second modification consisted of reduction in the layer thickness. The third modification combined the effects of the first and the second ones. For four out of five shapes tested the applied process modifications resulted in significant strengthening of the part. The shape that exhibited the best results was subject to further research by creating special printing mode. The mode included fine-tuning of three technological parameters on different stages of the part fabrication. As a result it was possible to increase the part strength by 108% only by tuning printing parameters of the best shape designed with increasing its weight by 8%.

## 1. Introduction

Additive manufacturing (AM) has been primarily considered as rapid prototyping technology for 10 to 15 years since its appearance [1,2,3,4,5,6,7] in the mid-1980s. In other words, AM was mostly used for creating mockups of new products or building functional prototypes. Since the mid-1990s, the rapid tooling concept started to emerge. Rapid tooling is an application of AM to produce tooling for conventional manufacturing processes such as casting, molding, and forming [8,9,10,11,12,13,14,15,16,17,18,19,20]. The next evolutionary step in AM has become rapid manufacturing: 3D printers are used for immediate manufacturing of the ultimate product [21,22,23,24] or spare parts [25]. In last several years the most prominent trend in AM development is increasing the availability and affordability of machines and supplies. There are relatively affordable solutions emerging for vat photopolymerization [26,27,28] as well as powder bed fusion [29], but the major share of affordable desktop machines operate on the material extrusion principle. The most widespread material extrusion AM technology is Fused Filament Fabrication (FFF). It uses polymer supplied in the form of filament that is melted, extruded, and deposited in layers. The FFF technology evolved fast thanks to the open source nature of the RepRap [30,31] project. Along with broad distribution and drastic price reduction of FFF machines and consumables, rapid manufacturing is now more and more often replaced with the term distributed manufacturing, implying the shift from making exclusive products to massive production of commodities [32,33,34,35,36,37,38]. Laying aside Fused Granular Fabrication (FGF) machines, operating on polymer pellets or regrind, as described by Woern et al. [39] and Reich et al. [40] showing the performance increased by 1200% as compared with a baseline of contemporary desktop 3D printer, the following tendency can be formulated. While the performance of machines printing with molten polymers has only increased a few times since their appearance, the average price of such a machine fell by several orders of magnitude. The relatively low performance of desktop 3D printers is compensated by their affordability and thus their broad distribution. Thus, the key advantage of the technology is flexibility rather than productivity. 

One of the most important factors for the success of the new manufacturing paradigm is the ability to ensure predictable quality when creating functional products. Thus, increasing the mechanical properties of the 3D printed products obtained with the FFF technology became an area of research and engineering of high interest. There are many approaches designed, many of them requiring hardware modifications in the 3D printers with FFF technology. Thus, Shih et al. [41] suggest low-temperature plasma to activate the sublayer, obtaining samples of double strength compared with reference. Go & Hart [42] proposed to preheat the filament with the laser, and Ravi et al. [43] used the laser to preheat the sublayer. Kishore et al. [44] heated the sublayer with infrared source. In experiment, performed by Lederle et al. [45] 3D printing is performed in the controlled atmosphere with oxygen excluded, resulting in a 38.6% increase in tensile strength for nylon samples (with relatively high standard deviation within the test group) and only 6.2% for ABS. There are also approaches introducing certain pre- or postprocessing applied to the raw material or to the printed part. For example, Abourayana et al. [46] utilized plasma treating for activation of pellets before the filament extrusion, thus increasing the tensile strength of the prints by 22% at the cost of multi-stage gas treatment process. Shaffer et al. [47] studied the influence of ionizing radiation on printed sample strength. A microwave heating is used for strengthening of printed parts in the research by Sweeney et al. [48], and ultrasonic vibrations are utilized for the same purpose by Li et al. [49], and by Tofangchi et al. [50]. Thermal annealing is applied to printed parts by Hart et al. [51] and Singh et al. [52]. There are recent studies of effects of new materials being included into FFF process, such as PC [53] or PEEK [54,55], as well as composites with discrete [56] and continuous fiber infill [57] suggested by Lu et al, graphene [58] by Caminero et al, and special additivities described by Levenhagen et al [59,60]. However, the potential to increase the strength of the samples printed with the most popular material, the Poly (Lactic Acid) or PLA, is not fully revealed. This increase in strength is possible by tuning printing process on a common desktop 3D printing machine.

It was shown in our previous work [61] that layer cohesion increases along with N/l value increase—the ratio of the diameter of the nozzle with which printing is performed to the layer thickness. It was also shown in another work [62] by our research team that temperature conditions at the point where a new layer is deposited also have a great influence on the formation of a durable product, as well as extrusion efficiency. Temperature conditions at the site of bond formation can be expressed through the sublayer temperature parameter. This is the temperature of the previous layer print, depending on the extrusion temperature, cooling conditions, and layer printing time (the latter, in turn, is determined by the print speed and part shape) at the moment when the same position in the new layer is extruded. Extrusion efficiency is defined as the ratio of the actual part mass to the calculated from models. The extrusion efficiency seems to be very dependent on hardware [63], but within single hardware configuration, it also depends on the melted polymer viscosity (which in turn depends on extrusion temperature) and the flow rate. 

The current study is the continuation of a recent study [64], which demonstrated the importance of part shape design with respect to FFF technology peculiarities and its influence on the sample strength. There were five different CAD models of an axisymmetric part with constant coupling dimensions converted into PLA FFF parts and then tested with radial loading (Figure 1). 

Shape 1 was used as a representative of common issue of FFF parts when the shell mimicking the part geometry interrupts forming a weak spot (Figure 2). Shapes 2–5 represent different attempts to solve the interrupted shell issue. Shape 2 is based on the traditional approach of increasing strength of weak sections by local addition of material (a fillet between shaft and boss section in this case). Shapes 3 and 4 are attempts to increase the part strength by having in mind specifics of FFF technology. While Shape 3 is formed on intuition, Shape 4 is a result of several iterations of design and simulations. Finally, Shape 5 is a combination of traditional approach (adding a fillet) and based on understanding FFF process design optimization (i.e., removing some CAD model volume to form continuous shell). By applying that combined approach it was possible to increase the part fracture load from 483 N (Shape 1) to 1096 N (Shape 5) with some reduction of weight of the part.

In the work [61] it is shown that model design is a keystone for making strong parts, but it lacks any effort to optimize printing parameters. The current paper aims to close that gap and assess the capabilities of different models designed to be printed with the highest possible mechanical properties.

The goal of current research is to establish more general guidelines to increase printed parts strength that would be applicable to FFF printers of different types. The current work shows the effect of tuning the FFF process parameters on the strength of the samples of the same five shapes. The tuning of process parameters is based on findings of previous works [61,62]. Along with “coarse” tuning—altering printing parameters for the whole printing cycle—“fine” tuning is also studied. In the latter case three parameters are varied during the printing cycle depending on the specific part of the sample being printed. It is shown that for a complex part, for an optimized geometry (and only for it), a significant increment of mechanical performance is achievable by optimization of FFF process parameters.

## 2. Methods and Materials

### 2.1. Sample Shapes

The shapes of samples tested were inherited from a previous study [64]. An item consisting of two connected coaxial cylinders of large (boss) and small (shaft) diameters was used as basic shape (Shape 1), the dimensions and constitution (superposition of shell, bases, and infill) of the printed part are shown on Figure 2. Four different approaches to increase the strength of such a part by modifying its shape and constitution resulted in four additional shapes (Shapes 2–5), its constitution is shown in Figure 3.

For CAD models with relatively large volume (Shapes 1 and 2), the configuration provided maximum strength in the previous study [64] (base thickness 1.2 mm, shell thickness 2.4 mm, infill 60%) was fabricated and tested. For CAD models with lower volume (Shapes 3–5), the same single configuration that was tested in the previous study [64] was used: shell thickness 1.8 mm, 100% infill, and no bases (base thickness 0 mm).

### 2.2. Samples Fabrication

A desktop Ultimaker 2 printer (Ultimaker B.V., Geldermalsen, The Netherlands) was used to produce all the samples. The specific machine used differs from the mass-market model with an installed alternative feed mechanism of the BondTech brand, built on a stepper motor with an integrated gearbox and drive to both feed rollers, and an alternative 3D Solex (Cepta AS, Oslo, Norway) heating unit with an increased power heating element (~50 W). The alternative heating unit, unlike the stock one, allows changing nozzles. In this series of experiments, a brass nozzle with a channel diameter of 0.6 mm was used instead of the 0.4 mm standard nozzle found in stock Ultimaker 2 printer. 

A turquoise PLA filament of 2.85 mm in diameter was used, produced by REC Company (Moscow, Russia). This specific manufacturer of filament was chosen due to locally produced material and the desire to obtain results comparable with previous studies [61,62,63,64].

The values of the following printing parameters remained constant during all experiments:Nozzle diameter (0.6 mm)Heated bed temperature (60 °C)The first layer thickness (0.3 mm)The first layer printing speed (25 mm/s)

In addition to the technological parameter combination tested in the previous study [64] (Mode A), four extra modes of the printing process were used (Table 1). Modes B–D were used for fabrication of samples of all the shapes considered. Mode E was developed and tested for the single shape (Shape 5), which demonstrated highest strength results in the previous study (printed in mode A). In modes A–D the printing parameters remained unchanged starting from the second layer until the end of fabrication. In E mode, the extrusion temperature, flow rate, and fan speed were changed during the printing process. 

For each observation mentioned in the work, five samples were made and tested. The paper presents the average values for each test lot, while the standard deviation is indicated after the average value in parentheses. The sample was placed at the center of the printer bed. The G-code file was prepared using Cura 15.02.1 software (slicer) [65]. All samples printed were weighed before mechanical testing using digital analytical scales ViBRA LF Series (Shinko Denshi Co. LTD, Tokyo, Japan). Measurement results were rounded to one decimal digit.

### 2.3. Mechanical Testing

Sample strength tests were carried out on a standard universal electromechanical testing machine IR 5057-50 (OOO Tochpribor, Ivanovo, Russia) with a digital control system. The samples were fixed with a specially designed and manufactured device (Figure 4). That fixture was mounted on a movable traverse of the testing machine. The top roller from the three-point bend test kit was used to apply radial load on the sample shaft. 

The tests were carried out at constant speed (10 mm/min) and continued until the sample was destroyed. During the tests displacements and loads were recorded. The reference point was the state of the machine with a load of 5 N applied to eliminate mounting clearances. The part strength was assumed to be equal to the load at which the first apparent crack appears. That point can be easily identified on the load–displacement curve. Along with absolute strength, relative strength (fracture load related to the sample mass) was also considered.

### 2.4. Sublayer Temperature Evaluation 

During the printing process, the temperature distribution over the surface of some samples was recorded using a FLIR B335 (FLIR Systems, Wilsonville, OR, USA) thermal imager with a resolution of 320 × 240 pixels. The camera was located in front of the printer at a distance of about 30 cm, which is equal to minimal focal distance documented for the imager. The software “FLIR Tools” [66] was used to determine the average temperature on the sample surface at a distance of approximately 1 to 3 mm from the lower cut of the nozzle (see Figure 5), this temperature was taken as the sublayer temperature (t_SUB_).

### 2.5. Macrostructure Evaluation of Fracture Surfaces

Images of fracture surfaces of tested samples were taken with a Sony A6000 (Sony Corp., Tokyo, Japan) digital camera, equipped with Sony E 35mm f1.8 lens and four macro rings with overall length of 52 mm. Images were taken with f/18 aperture, 3 to 5 sec exposure and ISO 100 sensitivity. For areas with deep relief several images with different focuses were taken and then combined with Enfuse 4.2 software [67]. Displaying the images on high definition computer monitor achieved approximately 100× magnification. The resulting images contained from 24.7 to 28 million pixels. 

## 3. Results and Discussion

### 3.1. Technological Parameters Coarse Optimization

The Table 2 shows the results obtained for all shapes tested in A–D printing modes. Data on A mode has already been published in recent study [64].

The results obtained for the Shape 1 samples are different from all the others: the proven recipes for increasing the interlayer bonding strength (tested when printing samples consisting of continuous shells) are completely inefficient for the interrupted shell case. All Shape 1 specimens tested were fractured at the boundary between the shaft and the base, same as in the previous study [64].

For Shapes 2–5, all optimized modes (B–D) tested showed significant increase in the part strength relative to the baseline (mode A). The maximum strength was recorded for parts printed in mode D, then modes B and C, in descending order. The sample mass changes were organized in descending order: D, B, C, and finally A.

Samples for Shapes 2, 4, and 5 printed in modes B–D were destroyed at greater load but at the same area as in the previous study [64] (corresponding to mode A). At the same time, qualitative differences occurred for Shape 3 samples destruction behavior (Figure 6). 

In samples printed on the modes A and C, the fracture occurred along the shaft section (Figure 6a). In samples printed in modes B and D the shaft remained solid and was pulled out from the boss (Figure 6b). In the second case, even after the cracks appeared and the shaft deviated from the original axis by up to 30°, the part did not break up into fragments. 

For all shapes printed in mode C, not only did the strength increase, but also an improvement in the visual surface quality (the resolution along the Z axis increases 3-fold) compared with the baseline (mode A), since surface roughness decreased together with layer thickness, and no distortions occurred. Modes featuring increased temperature and fans switched off (modes B and D), on the contrary, lead to distortions of the shape and defects on sample surface. The latter can be examined on the Shape 5 samples, the only model featuring horizontal overhanging elements. Printing such overhanging sections with no support is possible due to the “bridging” effect. Like a bridge span, portion of a certain layer is supported only by side pillars; in the case of Shape 5, these are outer rim and the central hub of the boss (see Figure 3d). The molten plastic threads stretch between the structures already printed as the nozzle moves between them. Sagging curvature depends on filament thread thickness, span length and plastic viscosity (extrusion temperature and cooling conditions). Increased extrusion temperature and disabled fans adversely affect the ability of the printer to make “bridges”. Figure 7 shows samples of Shape 5 below. 

In addition to sagging of unsupported areas, the “hot” settings (modes B and D) caused defects on the side surface of the shaft (Figure 8). 

These defects signify the overheating of the part during printing. The temperature of the upper printed layers depends not only on the extrusion temperature and cooling conditions, but also on the previous layers printing time. The latter, in turn, depends on the feed rate and the part shape. The upper layers temperature was recorded when printing in A–D modes (Figure 9) with the aid of infrared thermal visor. 

The highest temperatures were recorded when printing in mode B. When layers with large cross section (boss) have been being printed, the temperature remained at 60 °C, but as cross-section area decreased (shaft), the temperature gradually rose to 90, 117, and 122 °C. At temperatures exceeding the glass transition temperature (70 °C for PLA), the plastic remains pliable and easily deformed resulting in defects being formed (Figure 8). The slicer used in this study does not add a command to retract the filament at the moment of layer change, thus a small portion of excessive plastic is oozing and being added to the part while the table is moving down before the next layer is started. The build plate of the Ultimaker 2 is not rigid and is rested on three springs. If a nozzle collides with a minor flush on the sublayer, it is normally compensated by lowering of the build plate (with the springs being compressed underneath the plate). In case the sublayer becomes hot and thus soft, the nozzle just smashes a portion of already deposited plastic out of its way instead of pushing the part with the build plate down. These defects become especially noticeable when printing in mode D. Comparing modes B and D, mode B features slightly higher temperature of the upper layers, but for mode D, due to thinner layers, the hot nozzle moves much closer to the previous layer. 

Thus, any of the options considered for increasing the part strength by modifying the technological modes is a compromise. Decreasing the layer thickness leads to longer printing time, and increase in the sublayer temperature leads to the defects appearance on the surface. However, the negative consequences of such modification of print modes can be minimized. Printing with thinner layers (with relatively low flow rate) can be carried out at higher speeds, thus minimizing time loss. Overheating of parts can also be avoided if the fan blow rate is controlled depending on current layer area instead of simply disabling it.

### 3.2. Technological Parameters Fine Tuning

Part geometry optimization to increase its strength can be summarized, in general, to redistribution of material from less to more loaded areas. In the case of FFF technology, additional material can be supplied to the most critical elements of the part being printed without changing the CAD model. FFF is a technology where material is added layer by layer; thus, it is possible to add the material in larger or smaller quantities in different regions of the part. Material amount can be controlled through the flow rate parameter, available both in the slicer and through the printer menu. 

The default flow rate value, shown as 100% in printer menu, is calculated by the slicer using the formula
(1)$$Flow=N\times l\times F_{r},$$
where Flow is the flow rate (volumetric printing speed), mm^3^/s; N is the diameter of the nozzle channel, mm; l is the layer thickness, mm; and Fr is the feed rate (linear printing speed), mm/s.

The actual volume of material squeezed out by the correctly set up printer with a Bowden extruder (e.g., Ultimaker 2) through the nozzle per unit of time is always less than the calculated one: the slicer does not take into account the extrusion resistance [63]. As shown in previous study [62], even printing with plastic overheated to 250 °C at low speed (2.2 mm^3^/s) resulted in extrusion efficiency equal to 0.96 only. It seems that the only way to have extrusion efficiency equal to 1 or over 1 is to set the flow rate parameter above 100%. The Figure 10 shows the summary of a short experiment in extrusion efficiency test for the Ultimaker 2 3D printer. It used cylindrical specimens (D = 20 mm, h = 15 mm) printed in accordance with the settings of mode A and constituting of 1.8 mm shell and 100% infill with a flow rate from 100 to 130%. The calculated mass is 6.02 g, and the actual mass and diameter of the samples are given in the Table 3.

As it can be seen from the Table 3, an increase in flow rate of 10% still does not give an extrusion efficiency of 1 (measured by sample mass), but the resulting sample exceeds the diameter designed. An increase in Flow rate of 20 and 30% leads to significant distortion in shape, namely, to the appearance of a characteristic flush. The excessive plastic location is explained by asymmetrical cooling conditions typical for the Ultimaker printers—the nozzle is blown with two fans, while the left fan is about 30 mm closer to the nozzle than the right one. The part cools worse on the right side, shape loss occurs there. It should be noted that probably no 3D printer provides ideally uniform cooling.

In addition to the threat of defects appearance and deviation from nominal sizes, the increased flow (increased extrusion efficiency) can significantly increase the cohesion strength between the layers. In addition to the obvious increase in the contact area between the layers, the higher plastic flow leads to increased substrate temperature and pressure towards the sublayer. The maximum effect on particular part strength can be achieved by controlling the value of flow rate during the printing process. Along with this parameter, the other ones can be adjusted separately for different layers, for example, extrusion temperature and cooling conditions. 

An optimized printing script (mode E) was prepared for Shape 5 model, described in the Table 4. Parameter changes were made in the slicer using the Tweak At Z 4.0.1 plugin [68] for the Cura slicer. The same result can be achieved by manually editing the G-code or even by manually operating the printer during the printing process. 

Printing begins at standard extrusion temperature (210 °C) and continues until the overhanging part of the boss has finished printing. At the same time cooling intensity varies from 0 (printing the first layer when it is necessary to achieve high adhesion to the printer bed) to 100% (when the “bridge” is printed). This slice of the sample experiences relatively small loads; geometrical accuracy here is more important than strength. After the “flooring” of the overhanging part of the boss is complete, extrusion temperature is set to 250 °C, the fans are disabled, and flow rate is increased by 5%. In this way the loaded part of the boss is printed. The top layer of the boss is printed with even higher flow rate value (115%), thus forming dense foundation for the shaft. The shaft base is printed with the flow rate increased to 130%forming the critical area. To avoid substrate overheating and correlated defects, even before the critical zone is finished printing, the fans are turned on at the speed of 14% of the maximum. After printing the critical area of the part, the fans are accelerated to 24%, and flow rate is reduced to 100% in three stages. The samples obtained are thus without visible serious defects on the surface and have average mass of 34.2 (0.2) g. The average strength of the samples obtained under this scenario was 2139 (55) N. The average relative strength was 62.5 N/g. 

### 3.3. Fracture Surface Structure of the Tested Samples

Analyzing the structure of the surfaces exposed during the samples destruction during the tests allows one to better understand the nature of the processes and grasp the influence of 3D printing modes on the printed part properties. 

Comparison of the fracture surface structure formed during samples testing obtained in different modes using the example of Shape 5 is of particular interest (Figure 11). 

In all cases, the shaft destruction begins from above: the tensile stresses cause a crack to appear that quickly spreads downwards, completing the destruction. In the upper part of the shaft cross-section, signs of plastic deformation of the polymer can be observed. They appear in color photographs as whitish areas (Figure 11, top fragments). In the lower part of the cross-section, where fracture occurs by rapid crack growth, signs of plastic deformation are almost absent (Figure 11, bottom fragments). An exception is the cross-section on the specimen obtained in mode E: there are traces of plastic deformation on the entire surface of the fracture. The photos corresponding to different modes display visible quantitative and qualitative differences. There are the contours of the plastic threads forming the part clearly visible on Samples A and C. The samples printed in modes B and D have some fragments of the surface where the boundaries between the threads become indistinguishable. 

It is also interesting to analyze the crack surface structure of Shape 4 samples (Figure 12). The destruction is localized in the boss, where plastic threads of the infill predominate. As in the case described above, areas with plastic deformations are clearly readable as faded color of the plastic (white areas). It can be seen that the rupture, which was initially ductile in its nature, is rapidly spreading and destruction changes from ductile to brittle. By analyzing plastic deformations areas it can be concluded that a crack grows in two directions (Figure 12 and Figure 13). The crack spreads over the sample surface, gradually enclosing and dissecting the entire base of the shaft (blue arrows), and also penetrates deep into the part (red arrows).

The photos clearly show that even with 100% infill the real contact between the layers is dotted. The orthogonal threads of the adjacent layers have barrel-shaped cross-section and contact their neighbors in limited spots. Unlike the infill, the threads of the adjacent layers forming the shell are deposited parallel to each other. Thus, the layers do not contact in individual spots but in continuous strips (Figure 14). The latter can be clearly seen, for example, on photographs Shape 5 specimen fracture, obtained in modes A–C (see Figure 11). 

As the extrusion efficiency increases, the contact spots and stripes become larger. The figure shows that in modes B and D, the voids between individual threads are fragmented and turn into isolated caverns, approaching spherical shape. This effect was noticed and described in detail in previous work [62]. If the part heats up during the printing process and the temperature of the upper layers goes about 110–130 °C, at the moment of new layer deposition the plastic of the sublayer melts. The air filling discontinuities between adjacent threads is surrounded by liquid plastic and the surface tension forces contribute to the gradual transformation of the interface between air and plastic into the regular sphere. It is clearly visible that the air corridors at the boundaries between plastic threads are fragmented and coalesce on the fracture of the Shape 5 sample, printed in mode D. 

Finally, under the conditions of overextrusion, i.e., of an excessive plastic supply (mode E), the sample becomes almost solid—the boundaries between the threads become indistinguishable, air pockets are not formed. Any visible difference between the shell and the infill disappears.

### 3.4. Comparing Experimental Data with Previously Obtained Models

Controlling mechanical properties by modifying technological modes requires the result to be predictable. Previous studies derived regression models linking the printing modes (geometric [61] and temperature [62]) and part strength. The dependence of part strength from the N/l ratio is described by the formula
(2)$$Strength=70.7-{\left(\frac{N}{l}\right)}^{1.5}.$$


Accordingly, the predicted strength values for the modes A (N/l = 2) and C (N/l = 6) are 50.1 and 66.7 MPa, respectively. The predicted strengths bonus (the increase in strength for the transition from mode A to mode C) is 1.33. Table 5 shows the relationship of test results obtained for Shapes 2–5 printed in modes A and C. As it can be seen, the actual values of strength bonuses are quite close to the calculated value. Moreover, the average strength improvement for the four different shapes coincides with the predicted value with an accuracy of two decimal places—1.33. 

The model obtained in the previous study [61] is based on testing samples printed at a constant temperature (210 °C) and a constant feed rate (25 mm/s), it does not take into account the influence of thermal factors, and therefore is not applicable to assess the strength improvement, for example, caused by the transition from mode B to mode D. The model obtained in the other study [62], on the contrary, does not take into account geometric factors—it is completely built on samples printed with a single nozzle (0.6 mm) and layers of the same thickness (0.3 mm). The adequacy of this model can be assessed by comparing the test results for Shape 5 in modes A and B. The model relates the part strength with the extrusion efficiency value (E) and the sublayer temperature (t_SUB_):(3)$$Strength=23.945\times \ln\left(t_{sub}\right)+116.58\times E-153.912.$$

For Shape 5 samples, the calculated mass can be evaluated from the total filament length (4.55 m) consumed to print the product, calculated by the slicer and its density. One meter of the filament used weighs 7.92 g, thus the calculated mass of the Shape 5 sample is 36.0 g. Data on the sublayer temperature when printing the critical part (bottom shaft region) is shown in the Figure 9. Table 6 compares the predicted and observed parameters of the Shape 5 samples for modes A and B.

The calculated value of the strength increased caused by the transition from mode A to mode B is close to the actual. It can be stated that currently available models predict quite adequately the efficiency of strength improvement through printing mode optimization, however, these models are not complete. Additional experiments are needed to identify the joint influence of geometric and temperature factors on the part strength for desktop 3D printing technology. 

## 4. Summary

The effectiveness of coarse (modes B–D) and fine (mode E) FFF tuning for all tested shapes can be evaluated from Figure 15. Parts of Shape 1, containing critical shell interruption, cannot be strengthened by technological mode optimization as it is shown on the chart (red bars). For all other tested shapes modifying technological modes led to a significant positive effect. Significant increase in strength without loss of product surface and dimensional quality can be achieved by reducing the layer thickness (Shapes 2–5, mode C) or by fine-tuning the 3D printing parameters (Shape 5, mode E).

Comparison of the results obtained for Shape 5 using the A and E printing modes, shows that increasing sample mass by 8% caused strength improvement by 108%. This impressive result means, however, that in order to achieve best results, the attitude towards the 3D printer should be rethought. 

A desktop device operating on the FFF principle should not be viewed as a printer, that is, a tool for visualization (albeit in a tangible format) of computer models. It is rather a production machine with its own advantages and disadvantages, limitations, and capabilities. An ideal printer should have one single button that starts the process of turning a computer file into the printed counterpart. The user does not need to understand and to control the processes inside the black box. On the other hand, if the task is to get not “printouts”, i.e., mock-ups (cosmetic prototypes), but solid functional products, a production machine is needed rather than an end-user device. The ideal production machine allows the user to control and adjust all the technological modes. In this case, however, the user must be converted into a technologist, and the technological script for a particular product should be developed. Modern 3D printers are often subdivided into categories of affordable desktop machines (with cost up to USD 2000), much more expensive “professional” machines (with cost from USD 20000 upfront) and even more expensive “production” machines. In this sense, most devices in the category of expensive “professional” 3D printers remain “printers”. These machines are limited by a narrow range of proprietary materials and preset parameters values. On the other hand, open source low-end “printers”, being an order of magnitude more accessible, give the user full control over the technology. Transitioning from “rapid prototyping” to flexible manufacturing of functional things is possible on the existing hardware base of open source machines, in case product design is taken into account the features of technology and optimization of technological modes is done in accordance with the product shape and purpose.

## 5. Conclusions

To our best knowledge, the current study is the first description of effect of fine tuning the FFF process (the parameters were changed at specific layers) made with part geometry in mind. Fine tuning cooling conditions by changing fans rotation speed for different stages within single printing cycle allowed maintaining the sublayer to a temperature of approximately 90 °C, thus providing significant increase in interlayer bonding without causing any visible defects. Fine tuning extrusion temperature (up to 250 °C) and flow rate (up to 130%) allowed supply of additional material in the most critical section of the part and created an almost uniform and solid mesostructure of the part. Local increase of the flow rate, or controlled overextrusion, may be considered a unique option for the FFF process, not available to other AM technologies and processes without changing the CAD model. The fine tuning of FFF mode resulted in doubled part strength (the mechanical strength in a bending test rose from 1026 N, as achieved by design optimization in the previous study, to 2139 N) with negligible increase in sample weight (from 30.3 to 34.2 g). Generally speaking, fine-tuning the printing process parameters allows one to minimize negative and maximize positive effects. Meanwhile, technology optimization should be considered as the second step towards the launch of FFF printed product designed to bear loads. The first one remains product design optimization taking into account the features of the FFF technology. The study shows that printing mode optimization does not allow to increase the strength of a sample with a severe design flaw, in the case considered it was the interrupted shell occurring in the critical area (shape 1). Each and every manufacturing technology requires its peculiarities to be considered and process parameters tuned with regards to part shape and purpose. The attitude towards FFF printing process similar to any other production technology allows producing parts with high reliability features and predictable behavior under loads using desktop 3D printers.

## Figures and Tables

**Figure 1 materials-12-02071-f001:**
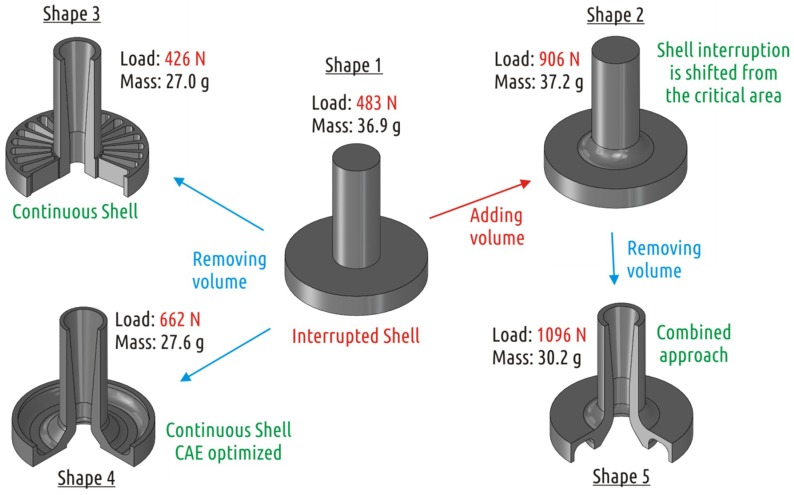
Testing part geometry optimization and results of the study [64].

**Figure 2 materials-12-02071-f002:**
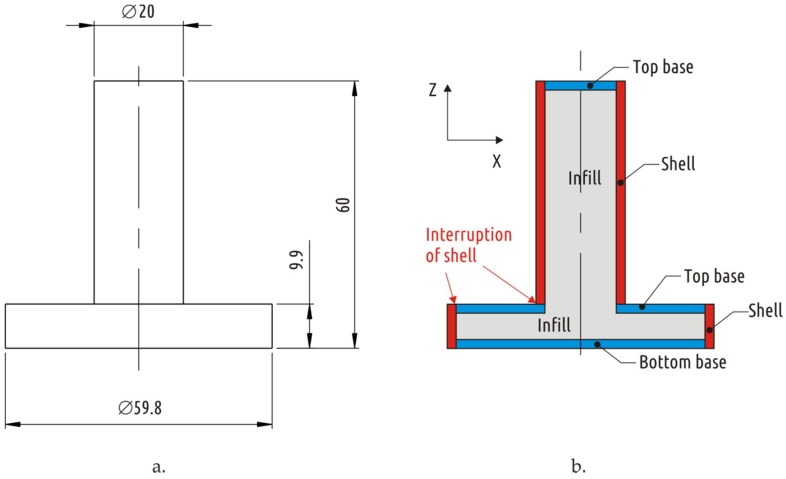
Shape 1 dimensions (unit: mm) (**a**) and constitution (**b**) with shell interruption highlighted.

**Figure 3 materials-12-02071-f003:**
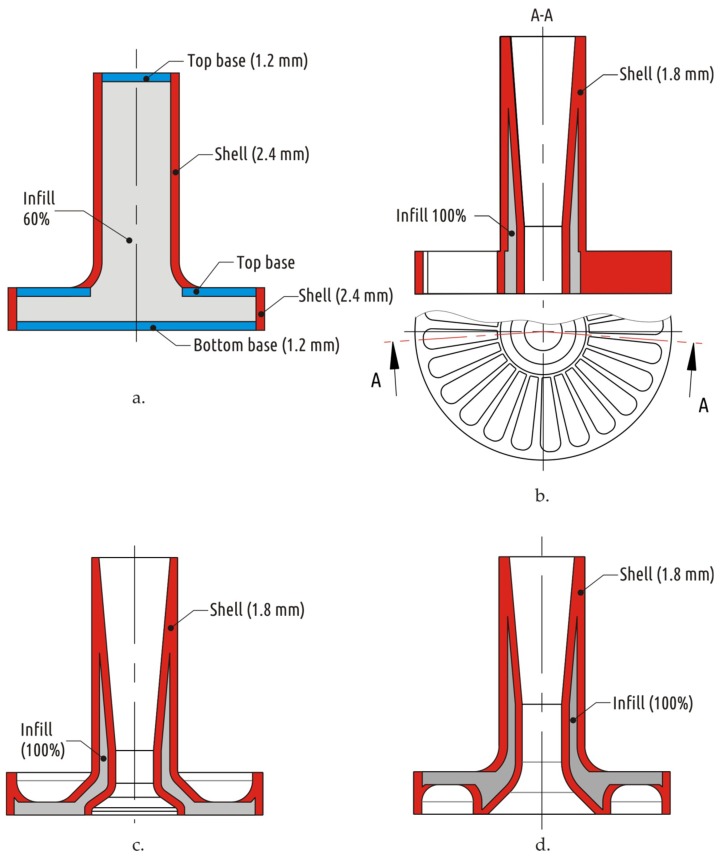
Constitutions of extra shapes, representing different approaches to strengthening the part: Shape 2—traditional approach (**a**); Shape 3—conversion of interrupted shell into continuous one (**b**); Shape 4—result of CAD/CAE iterative optimization (**c**); and Shape 5—combined approach (**d**).

**Figure 4 materials-12-02071-f004:**
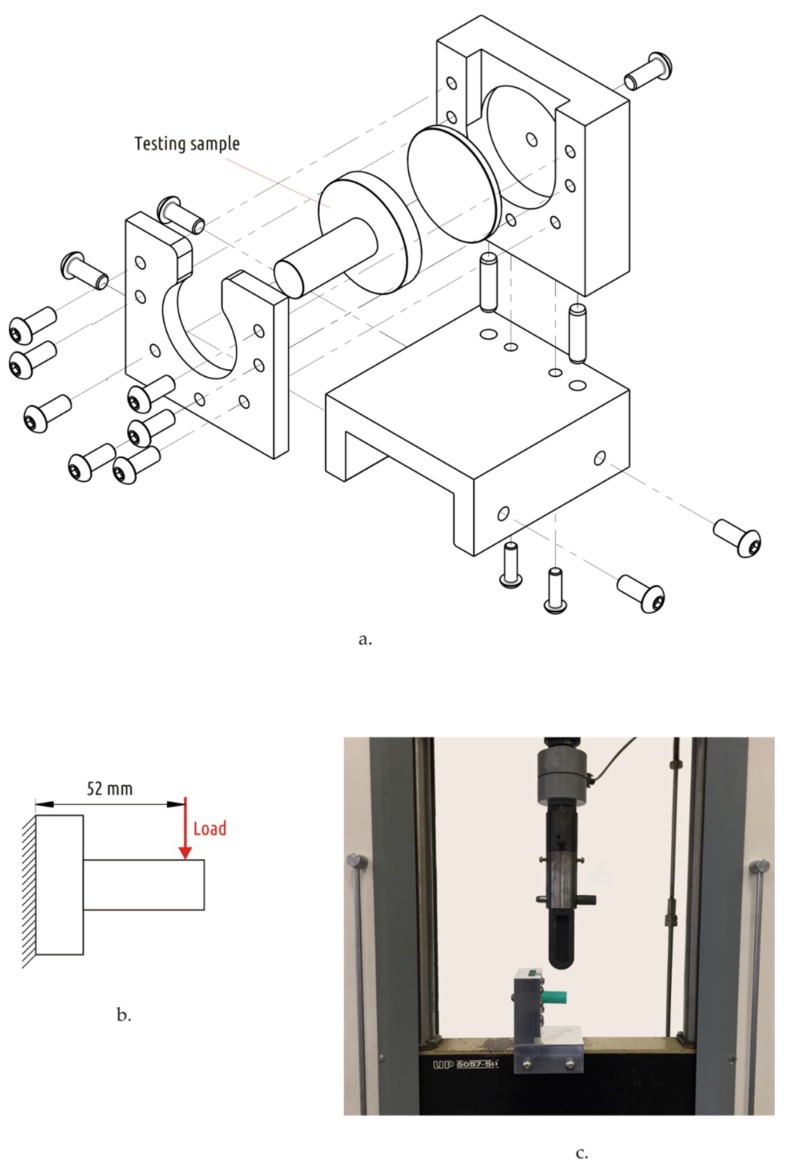
Sample fixture assembly (**a**), loading scheme (**b**), and overall view of testing apparatus (**c**).

**Figure 5 materials-12-02071-f005:**
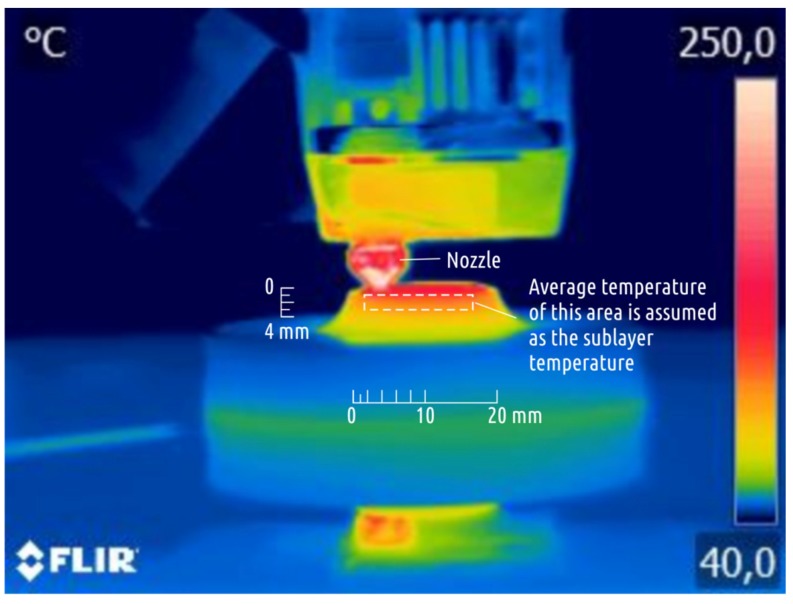
Sublayer temperature evaluation.

**Figure 6 materials-12-02071-f006:**
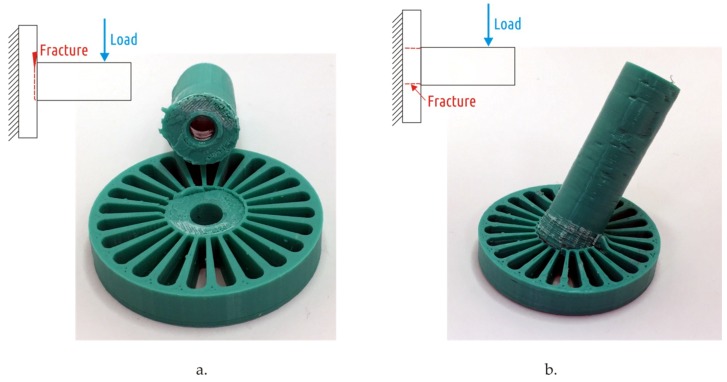
Destruction of Shape 3 samples printed in mode A [64] (**a**) and mode B (**b**). For the mode B sample, after the test is over, it is still not possible to separate the shaft from the boss with bare hands.

**Figure 7 materials-12-02071-f007:**
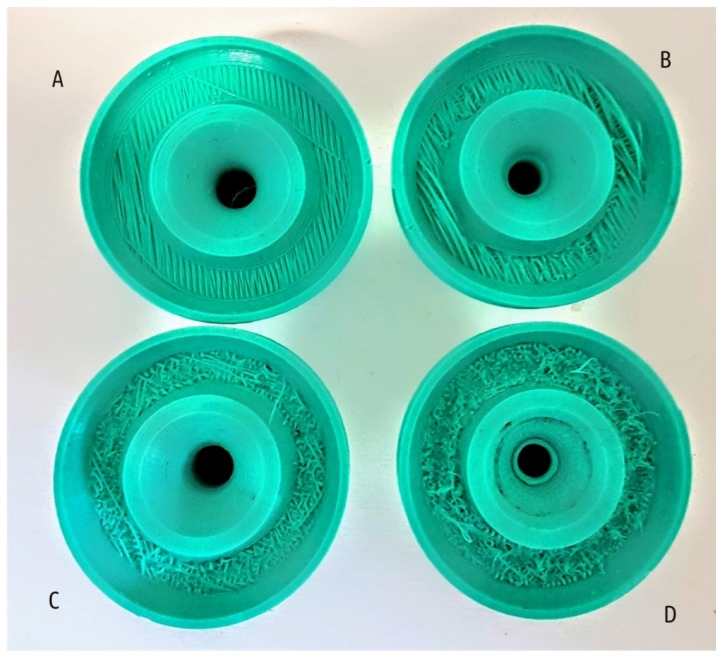
Shape 5 samples, bottom view: overhanging part of the boss when printing in the modes A, B, C, and D, as labeled.

**Figure 8 materials-12-02071-f008:**
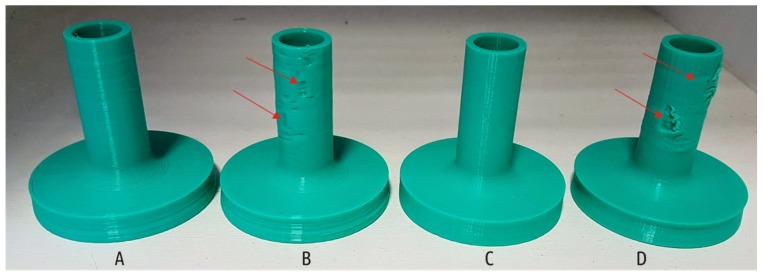
The appearance of the Shape 5 samples printed on the modes A, B, C, and D; red arrows indicate serious defects.

**Figure 9 materials-12-02071-f009:**
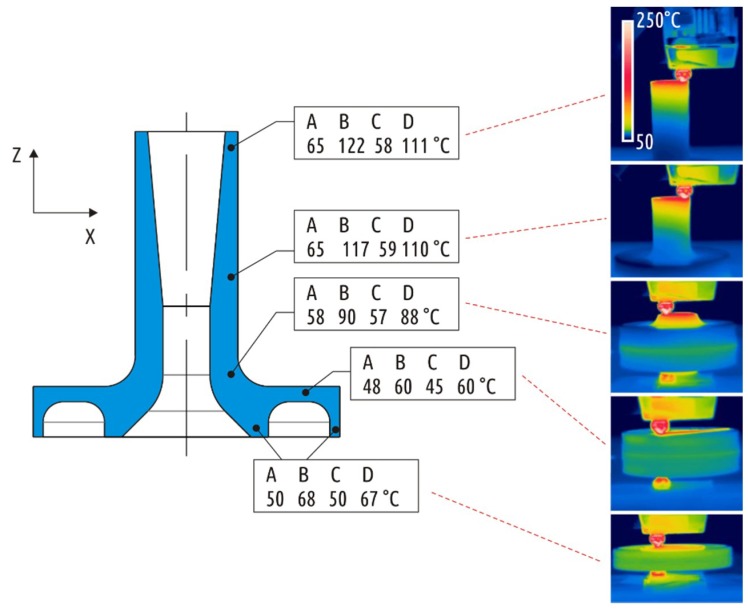
Sublayer temperature recorded at different stages of printing Shape 5 samples in modes A, B, C, and D (thermogram examples are given for mode B).

**Figure 10 materials-12-02071-f010:**
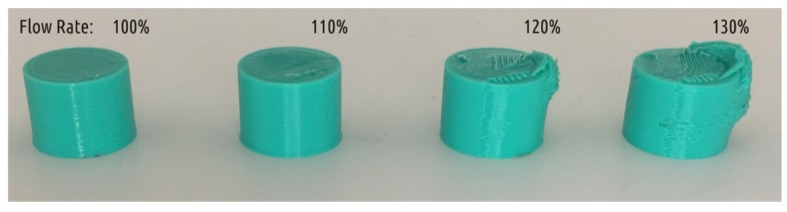
Cylindrical samples 3D printed with different flow rate settings, as labeled.

**Figure 11 materials-12-02071-f011:**
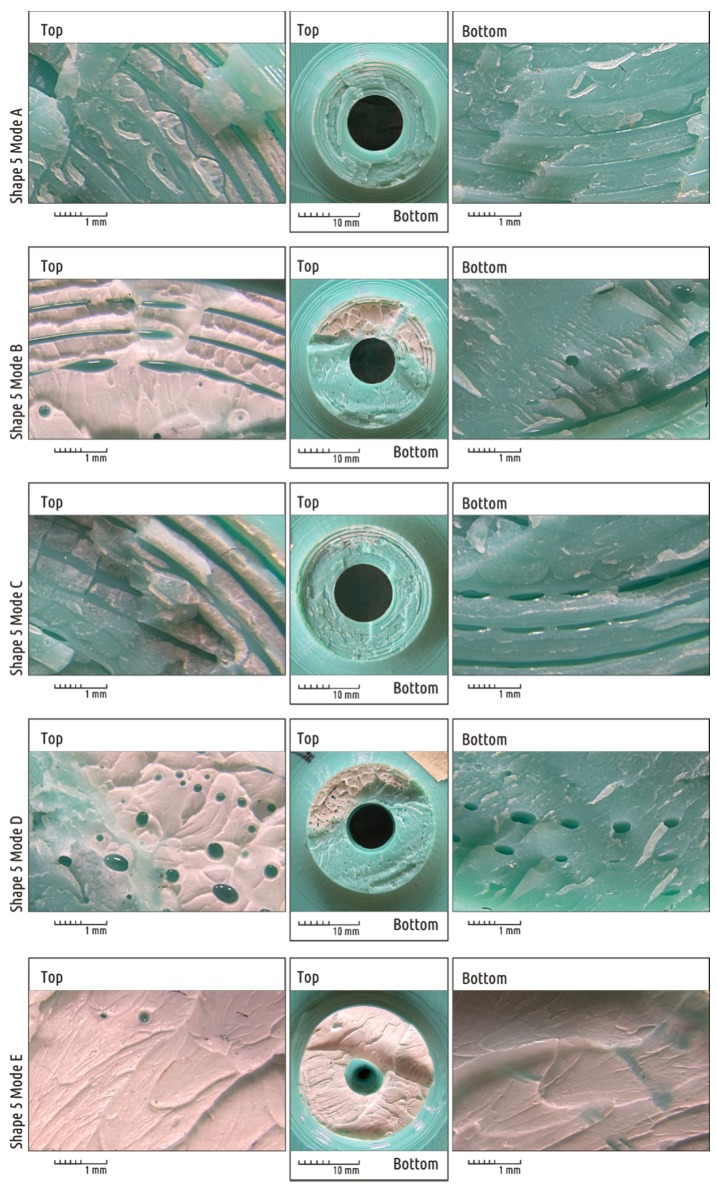
Fracture surface macrostructure of Shape 5 samples manufactured in different modes. “Top” and “Bottom” marks correspond to the position of the specimen at the time of destruction (the load was applied to the “top” surface of the shaft). An image of the entire fracture surface is given in the center for each sample. The fragments from the upper and lower parts are zoomed in to the left and to the right, respectively.

**Figure 12 materials-12-02071-f012:**
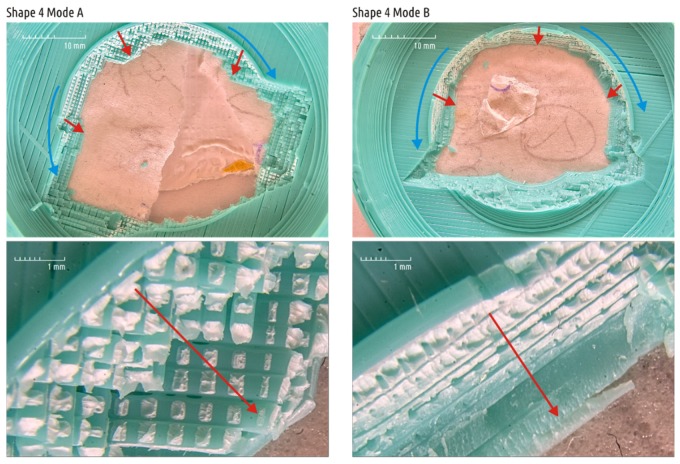
Fracture surface of Shape 4 samples obtained by modes A and B. The arrows indicate the direction of crack propagation.

**Figure 13 materials-12-02071-f013:**
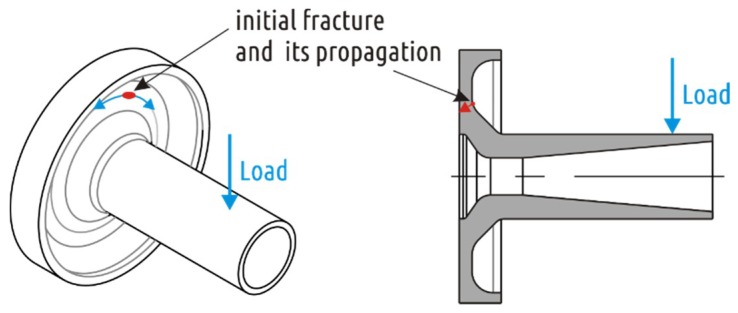
The initial rupture position on Shape 4 samples and the direction of crack propagation.

**Figure 14 materials-12-02071-f014:**
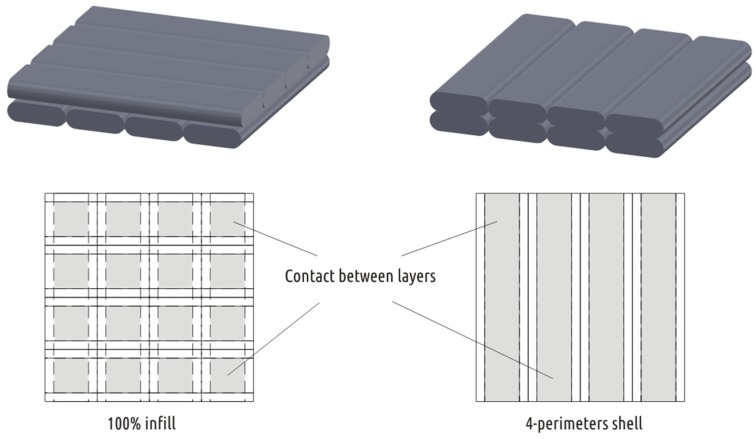
Superposition of polymer threads in adjacent layers of infill (left) and shell (right) and the contact area between the layers.

**Figure 15 materials-12-02071-f015:**
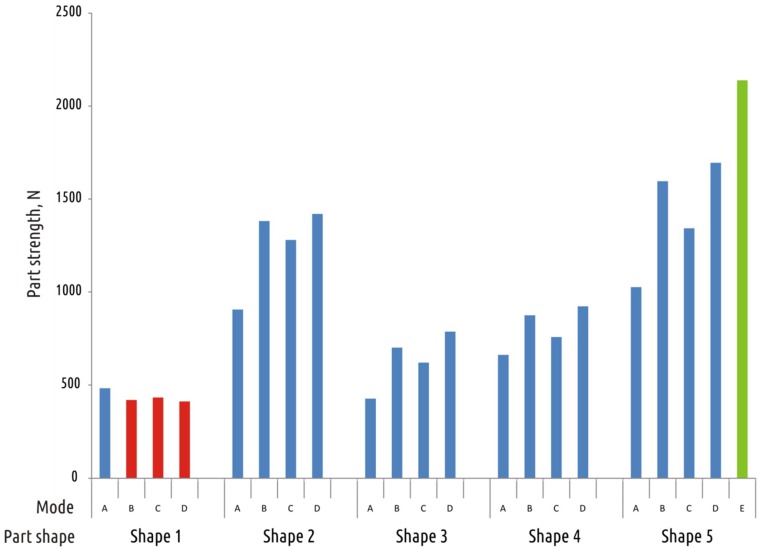
Summary on part strength achieved by different printing modes on all tested shapes.

**Table 1 materials-12-02071-t001:** Printing modes of samples fabrication.

Mode	Layer Thickness, mm	Print Speed, mm/s	Flow Rate, mm^3^/s	Extrusion Temperature, °C	Fan Speed, % *
A	0.3	30	5.4	210	100
B	0.3	30	5.4	250	0
C	0.1	60	3.6	210	100
D	0.1	60	3.6	250	0
E	0.3	30	5.4–7.02	210–250	0–100

* 0%—the fans are off; 100%—the fans are running at maximum speed.

**Table 2 materials-12-02071-t002:** Experimental data. The first value is the average and the value in parentheses denotes standard deviation.

Shape	Mode	Printing Time, hh:mm	Mass, g	Strength, N	Relative Strength, N/g
1	A	1:59	36.9 (0.2)	483 (47)	13.1
	B	2:01	37.5 (0.2)	420 (36)	11.2
	C	3:13	37.0 (0.2)	433 (32)	11.7
	D	3:14	37.6 (0.2)	412 (28)	10.9
2	A	2:01	37.2 (0.2)	906 (52)	24.4
	B	2:02	38.6 (0.2)	1382 (85)	35.8
	C	3:23	37.6 (0.1)	1280 (64)	34.0
	D	3:24	39.1 (0.2)	1420 (34)	36.3
3	A	1:48	27.0 (0.1)	427 (26)	15.8
	B	1:49	28.0 (0.1)	701 (55)	25.0
	C	3:34	27.7 (0.1)	620 (72)	22.4
	D	3:36	28.5 (0.1)	787 (44)	27.6
4	A	1:37	27.6 (0.1)	662 (51)	24.0
	B	1:39	28.7 (0.1)	875 (54)	30.5
	C	2:57	28.3 (0.1)	758 (21)	26.8
	D	2:58	29.1 (0.1)	923 (30)	31.7
5	A	1:46	30.3 (0.2)	1026 (72)	33.9
	B	1:47	30.9 (0.1)	1596 (32)	51.7
	C	3:18	30.8 (0.1)	1343 (46)	43.6
	D	3:18	31.6 (0.2)	1695 (68)	53.6

**Table 3 materials-12-02071-t003:** Short experiment on forced overflow data.

Flow Rate, %	Actual Mass, g	Extrusion Efficiency	Actual Diameter, mm
100	5.61 (0.1)	0.93	19.7 (0.1)
110	5.91 (0.1)	0.98	20.2 (0.1)
120	6.37 (0.1)	1.06	20.6 (0.1)*
130	6.87 (0.2)	1.14	20.7 (0.1)*

* Excluding flush.

**Table 4 materials-12-02071-t004:** Fine tuning of the Fused Filament Fabrication (FFF) process (mode E).

Layer #	t_E_, °C	Fan Speed, %	Flow rate, %	Comment
1–6	210	0	100	Initial printing on the hot plate
6–21	210	30	100	Printing the unstressed part of the boss
22–23	210	100	100	“Bridging”
24–31	250	0	105	Printing the stressed part of the boss
32	250	0	115	Top of the boss part
33–49	250	0	130	Printing the critical zone
50–69	250	14	130	Printing the critical zone, providing some cooling to avoid defects formation
70–99	250	14	120	Leaving the critical zone
100–129	250	24	110	Printing non critical part of the shaft
130–200	250	24	100	Printing non critical part of the shaft

**Table 5 materials-12-02071-t005:** Strengthening of parts by lowering the layer height (with transition from mode A to mode C).

Shape	2	3	4	5	Model
Mode A, strength, N	906 (20.1)	427 (14.2)	662 (18.4)	1026 (23.5)	-
Mode C, strength, N	1280 (22.6)	620 (19.3)	758 (18.8)	1343 (22.2)	-
Strength bonus	1.41	1.45	1.15	1.31	1.33

**Table 6 materials-12-02071-t006:** Strengthening of Shape 5 parts by increasing of sublayer temperature and extrusion efficiency (with transition from mode A to mode B).

Parameter	Shape 5, Mode A	Shape 5, Mode B
Calculated mass, g	36.0	
Actual mass, g	30.3 (0.2)	32.3 (0.2)
Extrusion efficiency	0.84	0.90
Sublayer temperature, ° C	58	90
Calculated strength, MPa	70.9 (0.9)	104.2 (1.1)
Actual sample strength, N	1026 (23.5)	1596 (31.2)
Estimated strength bonus	1.41	
Actual strength bonus	1.47	

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
