# Peer review of "Desktop Fabrication of Strong Poly (Lactic Acid) Parts: FFF Process Parameters Tuning"

_materials, 2019, doi:10.3390/ma12132071_

Round 1

Reviewer 1 Report

    This manuscript discussed the options to optimize the PLA parts printed on the desktop 3D printer from an Engineering perspective, which is the continuing study of the authors’ previous paper on Polymers. It brought up an excellent point to the future trend of the low-cost FFF printing, suggesting that equipping the open-source low-end 3D printers with the function of smart, automatic optimization of the part design before the printing is the way to improve the reliability of the parts manufactured from additive manufacturing. However there are several concerns as below that need the authors’ attention.

There are a lot more studies have been done on improving interlayer adhesion in the parts printed from low-cost FFF printers, which need to be included in the background section. Dr. Mark Dadmun’s group from UTK, and Dr. Jeffery Baur’s group from AFRL have done intensive research on this topic. Others, like Dr. Keng Hsu, etc. also have contributed to this area, who need to be recognized as well.

The authors did not the sample size for each of the printing conditions in the manuscript. If only one specimen per printing condition, the results are not convincing enough. 3 or more specimens are acceptable. If there were 3 or more specimens in the study, the results need to be presented as averages.

In Section 3.1, page 10, the authors claimed that all shapes printed in mode C had “an improvement in the sample surface quality” because the layer thickness was 1/3 of it in other modes. However, it is difficult to accept that the assumption that sample surface quality equals to how thin the layer thickness is. Sample surface quality includes the smoothness of the surface, which usually can be characterized from the surface roughness measurements.

In Section 3.1, page 11, the explanation the authors offered for the defects in parts printed from B and D in Figure 8 was the “collision between the moving nozzle and the deformed plastic of sublayer”. More in-depth or detailed explanations are needed here: how the sublayer deformed to start with? Theoretically, with the right Z axis control, the nozzle should not collide with the previous layers. The deformation was caused by the heat expansion of the PLA? Or the PLA sublayer was too soft to support itself? This needs to be clarified.

The authors did a good job in analyzing the pictures of fracture surface macrostructures, pointing out how the part failure started and propagated. However, some explanations are missing here, especially why some of the parts had the ductile deformation initially then changed to brittle destructions. This is critical because the material the authors chose, PLA, is usually semi-crystalline. The printing parameters fine tuning could also impact the crystallinity of the parts, which could further affect the interlayer adhesion. It would be helpful to the audience if the authors could provide some explanation or theories here.

In the results part, printing time for different shapes in different modes needs to be listed in the tables. Better quality of the printed parts could take longer time. This is important for the audience to know to balance the quality and the time for printing parts.

There are several grammar mistakes and figure number errors throughout the manuscript. The reference part was a mess: no unified format.

    This paper merits the publication on Materials once the above issues get addressed. It would have a good impact to the additive manufacturing industry if published.

Author Response

Dear respected Reviewer,
We are really thankful for the constructive advice and remarks. We tried to take all the suggestions into account and hope that the paper has significantly improved due to your help, both in terms of readability and academic writing.

Reviewer 2 Report

Dear Authors,

I have carefully read your draft paper and concluded that your study is useful and interesting and may be acceptable for publication after some major revisions. In this work little innovation is reported with respect to others carried out previously. Also, I have a few questions and concerns with your work as presented, which I invite the authors to address or explain, and which are detailed in the attached document.

Please note that the comments are intended merely to assist the authors in improving the paper and ensuring that published papers are of the highest quality. They are in NO WAY intended to discourage or demean the authors personally

Sincerely,

Reviewer.

Author Response

(The authors gave the same response as above.)

Reviewer 3 Report

The manuscript presents an interesting technique of tuning the FFF process parameter for obtaining a strong engineering part. The introduction provides preliminary background and motivation of the research using a desktop printer and corresponding test methods. The research design is well organized. There is a scope of improvement in the area of presenting experimental data. The results are not showing any statistical significance and reliability. 

Following is a list of specific comment and suggestions:

Keywords: "Mechanical Strength" is not used in the texts of the manuscript. Please explain "part/sample strength". 

The keyword “technology optimization” has been used nowhere in the text. Please use it for relevance.

Table 2: Please provide statistical significance and reliability of the result.

Line 318-319: "...the increased flow can significantly increase the cohesion strength between the layers." If it is true, then the part strength should drop with less flow (Mode A to C). The result is showing the reverse. Please explain.

Author Response

(The authors gave the same response as above.)

Reviewer 4 Report

Structure and chapters:

(1) Please connect better the literature review with the research gap. Explain why this work is relevant, and define why the part under study is relevant. Is it a spare part? if yes, you can mention it in the introduction.

(2) Finnish the introduction section in about line 97 after defining clearly what is the research gap and what is what you are doing.

(3) Starting from line 98 that part belongs to methodology. Please, remove it from the introduction and merge it with sub-section 2.1 sample shapes.

Terminology:

(1) You use the term FFF to describe "Material Extrusion" technology. I would advise you to use standard terminology to describe AM technologies. Replace FFF term, or at least mention at the beginning of the manuscript that FFF belongs to the technology category of material extrusion. See:

ISO / ASTM International (2015) ‘ISO / ASTM52900 - 15: Standard Terminology for Additive Manufacturing – General Principles – Terminology’. doi: 10.1520/F2792-12A.2.)

(2) There are some typos and mistakes when referring to FFF, see line 75 and correct. Check the overall document for consistency.

(3) I would use the term 3D printing to refer to the overall technology. Instead use AM. See line 83, 85, etc and consistently change the term to AM and Material extrusion. Check the overall document for consistency.

(4) Why you call it "technological modes" instead of "process parameters". I recommend reviewing and replacing this term in the manuscript. It is somehow misleading and "process parameters" is more accurate and widely used to describe what you are doing.

Literature flaws:

(1) In line 73 you mention the transition from Rapid Manufacturing to Distributed manufacturing. Is this entirely true? Consider reviewing your argument. AM has still a way to go before can be used the way you describe.

See :

Flores Ituarte, I., Khajavi, S. H. and Partanen, J. (2016) ‘Challenges to implementing additive manufacturing in globalised production environments’, Int. J. Collaborative Enterprise, 5(3/4), pp. 232–247.

(2) Can you refer for example to a more common application of AM related to the replacement of legacy parts and spare-part application? Material extrusion is heavily used in these industrial applications, and redesigning components like the one to present can help industrialize the technology.

See:

Ballardini, R. M., et al. (2018) ‘Printing spare parts through additive manufacturing: legal and digital business challenges’, Journal of Manufacturing Technology Management, 29(6), pp. 958–982. doi: http://dx.doi.org/10.1108/MRR-09-2015-0216.

(3) In line 89 you refer to post-processing. Can you refer to literature and discuss why is used?. (For example to increased dimensional accuracy, preparation of AM for assembly, etc)

See:

Jasgurpreet Singh Chohan, Rupinder Singh, (2017) "Pre and post processing techniques to improve surface characteristics of FDM parts: a state of art review and future applications", Rapid Prototyping Journal, Vol. 23 Issue: 3, pp.495-513, https://doi.org/10.1108/RPJ-05-2015-0059

Methodology:

(1) You can consider including and schematic showing all the logical step in your methodology in a diagram. This can go at the beginning of the methodology section. It can help to add clarity to the manuscript and help the reader to follow the logic.

Author Response

(The authors gave the same response as above.)

Round 2

Reviewer 1 Report

The revised manuscript has addressed the reviewers' concerns and is suitable to be published on Materials.

Author Response

Thank you very much, we appreciate all your suggestions. 

Reviewer 2 Report

Dear authors,

You have have taken into account the recommendations of all reviewers, and now the article looks better.

Although I think it should be published in another journal such as Applied Science, I think that if the editor deems it convenient, it may be of interest to some authors of this journal.

Sincerely,

The reviewer

Author Response

Thank you, your point is appreciated. We would prefer to stay with the Materials, but, now it is the editorial decision.